# EBV Impact in Peripheral Macrophages’ Polarization Cytokines in Pediatric Patients

**DOI:** 10.3390/v15102105

**Published:** 2023-10-17

**Authors:** Agustina Moyano, Natalia Ferressini Gerpe, Maria Eugenia Amarillo, Elena De Matteo, Maria Victoria Preciado, Maria Soledad Caldirola, Paola Chabay

**Affiliations:** Multidisciplinary Institute for Investigation in Pediatric Pathologies (IMIPP), CONICET-GCBA, Molecular Biology Laboratory, Pathology Division, Ricardo Gutiérrez Children’s Hospital, Buenos Aires 1425, Argentinamaria.eugenia.amarillo41@gmail.com (M.E.A.); preciado@conicet.gov.ar (M.V.P.); mariasoledadcaldirola@gmail.com (M.S.C.); pchabay@conicet.gov.ar (P.C.)

**Keywords:** macrophages, children, EBV, cytokines, tonsil, periphery

## Abstract

Macrophages are exceptionally flexible cells. The presence of inflammatory cytokines such as IFN-γ and TNF-α results in an M1 (CD68) activation, while cytokines such as IL-10 or TGF-β induce the M2 (CD163) activation. Our aim was to study the behavior of peripheral cytokines involved in macrophage polarization and relate them with tissue findings to further comprehend the role of macrophages in EBV pediatric infection. We studied cytokine expression in tonsils and peripheral blood samples of children in different stages of infection. Peripheral cytokines were compared with macrophage polarization markers and viral protein expression in tonsils. Only IL-10 showed a negative correlation between compartments, exclusively in patients undergoing viral reactivation (R). Higher expressions of peripheral IL-1β, IL-23, and IL-12p40 in R children were observed. Lower expressions of local and peripheral TNF-α in patients with broader expressions of latent and lytic viral proteins were demonstrated. In healthy carrier (HC) patients, IL-23 positively correlated with CD163, and IP-10 positively correlated with CD68. Our results indicated that EBV might modulate antigen expression in the presence of TNF-α and influence peripheral cytokine expression differently in each stage of infection. Moreover, peripheral cytokines might have a particular role in macrophage polarization in HC.

## 1. Introduction

Epstein–Barr virus (EBV) is a gamma herpesvirus ubiquitously distributed worldwide that silently infects more than 90 percent of the human population. It disseminates through saliva and reaches the B lymphocyte, its target cell, in the tonsil. EBV was the first human virus to be associated with tumorigenesis. As all herpesviruses, it establishes two life cycles: a latent infection phase, during which the virus persists silently as an episome in the host cell, and a lytic replication phase, which allows the virus to spread [1]. In each cycle, viral protein expression differs, and many of them have been described to have oncogenic functions. The EBNA2 latent antigen is essential for the growth transformation of lymphocytes and is also a transcriptional transactivator [2]. Moreover, LMP1, which is considered the EBV’s most relevant latent oncogenic protein, has profound effects on cellular gene expression and induces multiple genes, such as adhesion molecules and growth factors, among others [3]. LMP1 is consistently expressed in Hodgkin Reed–Sternberg cells (HRS) in all EBV-associated Hodgkin lymphoma (HL) cases and is responsible for the induction of cytokines and chemokines that trigger the cHL hallmark inflammatory reaction [4]. There are numerous studies where EBV’s oncogenic capacity is evidenced, and it has been associated with a large variety of tumors [5].

Interestingly, our group previously demonstrated that in Argentina, different pediatric lymphomas associated with EBV (Hodgkin, Burkitt, and diffuse large B cell lymphoma) presented higher prevalence in children younger than 10 years old [6]. The age of primary infection (PI) differs between rich vs poor-resource countries. In the first ones, it occurs in the bimodal peak, and when the infection occurs in adolescence, it generally presents as an acute, self-limiting febrile illness characterized by the development of lymphadenopathy, known as infectious mononucleosis (IM). In the latter, it occurs at younger ages and is asymptomatic [7]. This distinct behavior may lie in the immune response to EBV that widely differs between pediatric and adult patients. Adults present a widespread proliferation of activated CD8+ T cells and little, if any, expansion of the CD4+ compartment responsible for the clinical manifestations. Conversely, in children, even though the existence of EBV-specific CTL has been described [8], there is no remarkable expansion of the peripheral lymphocyte subset, and it has been hypothesized that the innate immune system cells may play a key role in preventing symptom development [8,9]. NK cells are the main actors in the context of innate responses against EBV, particularly in children, and they are thought to have a key role in restricting EBV infection symptoms and EBV-mediated transformation, mostly via IFN-γ [10,11].

Dendritic cells can sense EBV via toll-like receptor 3 (TLR3) and TLR9, whereas monocytes and monocyte-derived macrophages have been described to sense EBV also via TLRs, especially TLR2 and TLR9. In addition, EBV-encoded secreted BARF1 (sBARF1) was found to inhibit M-CSF, restraining monocyte differentiation and macrophage function [12]. Our group previously reported a characteristic recruitment of macrophages surrounding EBV+ cells in tonsils [13], as well as a particular recruitment of alternative activated (M2) macrophages in pediatric patients with broader expressions of latency antigens [14]. Other than these findings, little is known about the macrophage’s role in EBV infection. Macrophages originate from peripheral blood monocytes attracted by chemokines and cytokines to the tissue. Their intrinsic heterogeneity during differentiation originates from reciprocal interactions with neighboring cells, including macrophages themselves, microorganisms, sterile particulates, and soluble mediators [15]. Originally, two polarization profiles were defined: the classically activated macrophages (M1) and the alternatively activated macrophages (M2). M1 polarization is induced by TNF-α, IFN-γ, IL-23, and bacterial components, such as LPS, and displays a high expression of CD68. These macrophages have high phagocytic capacity and are potent pro-inflammatory effectors that release high levels of cytokines, such as TNF-α, IL-6, IL-1β, and IL-12, but they also have a large antigen presentation capacity to promote Th1 response [16]. Therefore, they play an important role in pathogen and tumor cell control and elimination [17,18]. In contrast, M2 macrophage activation occurs in response to specific IL-4, IL-10, IL-13, or TGF-β cytokine stimulus. These macrophages express high surface levels of CD163 and scavenger receptors involved in debris clearance [19]. They display a tolerogenic phenotype characterized by decreased antigen presentation and the production of cytokines, such as CCL17 (TARC) and IL-1Ra (IL-1 receptor antagonist), that stimulate a Th2 response and originate a tumorigenic propitious environment [18,20]. In addition to this coarse classification, due to their high plasticity, intermediate phenotypes exist between M1 and M2 profiles [21]. Another particular macrophage phenotype is the one described in tumor microenvironments (TME), known as tumor-associated macrophages (TAM). These macrophages share several functionalities with the M2 profile and have an active role in tumor development and progression [22]. They secrete chemotactic mediators, such as CCL2, CCL3, CCL17, CCL18, and CCL22, that trigger the migration of functionally suppressive regulatory T cells [23]. In addition, they are involved in the regulation of protein expression, such as programmed cell death protein 1 (PD1) and its ligand PD-L1, which induce exhaustion and immune tolerance in T cells. In fact, in adult HL, PD-L1 in the TME was described to be mostly expressed by TAMs [24]. Furthermore, our group evidenced that latent and lytic viral proteins have influence on PD-L1 expression in macrophages in pediatric patients infected with EBV [25]. These findings prompted us to hypothesize that macrophages might have a key role in EBV infection control and hence viral contribution to lymphomagenesis. In pediatric patients in particular, the immune response against the virus is mostly mediated by NK cells as part of the innate immune system [26]. In addition, macrophages play a crucial role in tumor progression, and their great plasticity regarding different stimuli in the microenvironment has been widely described [17,18,20]. Understanding the interplay of macrophages and EBV in pediatric patients’ infections may help clarify viral contribution to the process of EBV-associated lymphomagenesis in children. In a previous study, our group characterized the macrophages’ polarization in EBV+ pediatric patients’ tonsils, viral entrance, and the replication site [14]. In this work, our aim was to evaluate the behavior of peripheral polarization cytokines and their relationship with tissue findings to further comprehend the role of macrophages in EBV infection and their potential contribution to lymphomagenesis.

## 2. Materials and Methods

### 2.1. Patients and Samples

A cohort of 59 patients, aged between 1 and 15 years (median age of 5 years) undergoing tonsillectomy due to nonreactive hyperplasia at the Otorhinolaryngology Division, Ricardo Gutierrez Children’s Hospital, were analyzed. Fresh tissue samples and a concomitant blood sample were collected. A portion of the tissue was preserved at −70 °C for nucleic acid extraction, and the remaining biopsy tissue was formalin-fixed paraffin-embedded (FFPE) at the Pathology Division of Ricardo Gutierrez Children’s Hospital. From blood samples, serum was separated, aliquoted, and preserved at −20 °C for later analysis. A fraction of serum was used to determine the EBV infection status by indirect immunofluorescence assay, as previously reported [27] (Appendix A). The evaluated cohort in this study was part of a larger cohort previously analyzed [14] and included 20 primary infected (PI) patients, 23 healthy carrier (HC) patients, 11 patients undergoing reactivations (R), and 5 not-infected (NI) patients (Appendix A).

### 2.2. Immunohistochemistry

Immunohistochemistry (IHC) was performed on serial cuts of the FFPE biopsies (3–4 µm). For viral antigens, LMP1 (CS1-4 pool of clones, Dako, Santa Clara, CA, United States) and EBNA2 (1E6 y R3 clones, Abcam, Cambridge, UK) primary antibodies as latency antigens and BMRF1 primary antibody (G3-E31 clone, Abcam, Cambridge, UK) as the early lytic antigen were used as described [27]. Cytokine detection was assessed with specific primary antibodies against IFN-γ (polyclonal, Abcam, Cambridge, UK), IL-10 (Ab34843, Abcam, Cambridge, UK), and TGF-β (Ab9758, Abcam, Cambridge, UK), as described previously [14]. TARC (CCL17) chemokine IHC was performed using Tris-EDTA buffer pH = 9.0 for antigen retrieval. After incubation with the primary antibody, immunodetection was performed using a commercial kit (ScyTek), followed by diaminobenzidine (DAB) as a chromogen. Lastly, macrophage polarization markers were accomplished using specific antibodies, CD68 (clone KP-1) and CD163 (clone MRQ-26), from Roche Ventana (Tucson, AZ, USA). The M1 profile was defined as CD68/CD163 > 1.5 and M2 as CD163/CD68 > 1.5, as described previously [14]. Stainings were evaluated by differentiating two tonsils’ histological regions. To better represent the overall tissue, for each patient, 10 pictures of the interfollicular region (IF) and 10 of the germinal center (GC) were taken at 400× and 1000× using the ZEN 2 (blue edition) imaging platform, with A1 AxioScope and a Carl Zeiss microscope. For cytokines (IL-10, IFN-γ, and TGF-β), TARC, and macrophage polarization markers (CD68 and CD163), positive cell counts were performed, and results were expressed as positive cells per area unit (cells+/mm^2^). For viral proteins, cell count was assessed in the whole tissue section and expressed as positive cells per area unit (cells+/mm^2^) as well. The mean cell count for each viral protein was 14.98 cells/cm^2^ for LMP1, 6.64 cells/cm^2^ for EBNA2, and 2.52 cells/cm^2^ for BMRF1 (Appendix A).

### 2.3. Real-Time PCRA

To complete the cytokine expression panel, IL-1β and TNF-α were evaluated by RT-PCR, as previously described [14].

### 2.4. Flow Cytometry

To analyze IL-12p70, TNF-α, IL-6, IL-10, IL-1β, TARC (CCL17), IL-1Ra, IL-12p40, IL-23, and IP-10 polarization, cytokines in the patient’s serum were tested using a fluorescence-encoded bead-based multiplex assay panel (Human M1/M2 Macrophage Panel, 10-plex, LEGENDplex, BioLegend, San Diego, CA, United States), as indicated by the manufacturer. Cells were acquired in a FACSCanto II flow cytometer (BD), and results were analyzed using a specific program provided by BioLegend.

### 2.5. Statistical Analysis

The data was analyzed using GraphPad Prism 8 software. A normality test was performed using the Shapiro–Wilks test. The comparison differences between the two groups were assessed using a *t*-test or Mann–Whitney (M-W), according to normality results. The comparison between groups was assessed by one-way ANOVA or Kruskal–Wallis tests, according to normality test results. Correlations were performed using Spearman or Pearson tests. Outliers were defined using the robust test to compare data median absolute deviation (Mad) in Excel. All tests were two-tailed, and *p* < 0.05 was considered statistically significant. Heat map and correlation matrix graphs were obtained with MetaboAnalyst 5.0.

## 3. Results

### 3.1. Peripheral Cytokines

A correlation matrix was carried out between the peripheral cytokines in order to evaluate cytokine behavior, and two well-differentiated groups were evidenced. Notably, IL-1RA was clustered with pro-inflammatory IL-12p70, IL-12p40, and IL-23. The second cluster was compounded by IL-10 and IL-1β and IL-6, TNF-α, CCL17 (TARC), and IP-10 (Figure 1A). In order to evaluate the cytokine behavior regarding infectious status and macrophages’ polarization profiles, a heat map arranged by both conditions individually was performed, and a small differential cytokine expression between PI, HC, and R patients was observed (Figure 1B). No specific patterns for polarization markers and cytokine expression were demonstrated. Lastly, in order to evaluate if the ages of our patients had any impact on peripheral cytokine expression, given that the mean age of our cohort was 5 years, we compared each cytokine expression between patients younger and older than 5 years. No significant differences were observed (*p* > 0.05; M-W). However, when comparisons for cytokine expression were settled in children younger and older than 10 years, IL-12p70, IL-1RA (IL-1 receptor antagonist), and IL-23 were higher in the elder group (*p* = 0.0061; M-W) (*p* = 0.0223; M-W) (*p* = 0.0230; M-W). This group (older than 10) included only 10 patients, among whom the four stages of infection and viral latency proteins were represented.

### 3.2. Peripheral and Tissue Cytokines

In many pathologies, the peripheral compartment reflects tissue events and is used for disease monitoring, due to its easy access. Here, we evaluated if there was a connection between peripheral and tissue cytokine expressions (Appendix A) in different infection statuses. No correlation was observed for TNF-α, IL-1β, CCL17, or IL-10 expression in tonsil and peripheral when the cohort was analyzed as a whole (*p* > 0.05, Pearson and Spearman). Conversely, when each infection subgroup was separately analyzed (PI, HC, R and NI), IL-10 showed a negative correlation between compartments (r = −0.6598; *p* = 0.0272, Pearson) only in R patients (Figure 2A). CCL17 cell count, assessed by IHC in tonsils, was statistically higher in the GC region in the entire cohort (*p* < 0.0001, *t* test), PI (*p* = 0.0002, *t* test), HC (*p* < 0.0001, *t* test), and R (*p* = 0.0022, *t* test) (Figure 2B). Therefore, the CCL17 expressed at GC was considered for further analysis.

### 3.3. Peripheral Cytokines among EBV Infection Status

In order to explore the differences in peripheral polarization cytokines in different EBV infection statuses, mean values for IL-12p70, TNF-α, IL-6, IL-10, IL-1β, CCL17, IL-1Ra, IL-12p40, IL-23, and IP-10 were compared among them. No significant differences were observed in serum cytokine expressions between EBV+ and EBV− patients (*p* > 0.05 *t* test and M-W), except for CCL17, which was higher in NI patients (*p* = 0.0267, M-W). Among infection status (PI, HC and R), only IL-1β (*p* = 0.0384, *t* test), IL-12p40 (*p* = 0.0272, *t* test), and IL-23 (*p* = 0.0319, M-W) presented significant differences between R and HC, with R being higher (Figure 3).

### 3.4. Peripheral Cytokines and Viral Protein Expression

As EBV latent and lytic proteins have been demonstrated to modulate intracellular pathways in different cells, viral protein expression was evaluated to investigate if latent and lytic antigens in the tonsils had any relation to polarization cytokine expression at the periphery. At first, patients were clustered into two groups: those who expressed Latency 0 and Latency I antigens as the L0/-I group and patients expressing Latency II and Latency III antigens as the LII/III group (Appendix A). Both peripheral IL-1Ra (*p* = 0.0242, M-W) and TNF-α (*p* = 0.016, M-W) presented higher expressions in patients expressing L0/I antigens (Figure 4A). Thereafter, we compared serum cytokine quantifications in the presence and absence of the BMRF1 lytic viral protein and remarkably, a higher expression of TNF-α in the absence of the lytic antigen (*p* = 0.0174, M-W) was demonstrated (Figure 4B). No correlations between peripheral cytokines and viral antigen cell counts in the tonsil were demonstrated.

### 3.5. Peripheral Cytokines and Macrophages’ Polarization

Due to macrophages’ intrinsic heterogeneity, the interactions with cells and soluble mediators at the surrounding environment are decisive for the polarization direction. In a previous work, we studied the macrophage polarization environment of EBV-infected patients’ tonsils, describing some interesting findings, such as the prevalence of M1 macrophage polarization, even in the presence of immunomodulatory cytokines [14]. To further explore if macrophage polarization at the tissue mirrored the scenario in the periphery, the correlation between each plasmatic cytokine (IL-12p70, TNF-α, IL-6, IL-10, IL-1β, CCL17, IL-1Ra, IL-12p40, IL-23, and IP-10) and CD68 and CD163 cell counts as M1 and M2 polarization markers, respectively, in the entire cohort and within subgroups (PI, HC, R and NI) was evaluated. IL-23 revealed a positive correlation with CD163 only in HC patients (r = 0.432; *p* = 0.0392, Pearson), and IP-10 positively correlated with CD68 only in HC as well (r = 0.4781; *p* = 0.0284, Spearman). Finally, we compared cytokine behavior as the mean value for each cytokine between M1 and M2 patients (Figure 5); while only CCL17 revealed a broader presence in M2-polarized cases (*p* = 0.0123, M-W), surprisingly, no differences were observed for the remaining cytokines (*p* > 0.05, M-W). Given that plasmatic CCL17 prevailed in polarized M2 tonsillar macrophages, and CCL17 cell numbers prevailed at the GC, the number of CCL17+ cells were correlated with polarization markers. A trend for a negative correlation between CCL17 and CD163 in HC (*p* = 0.0515; r = −0.4203) and a statistical negative correlation in the GC of R patients (*p* = 0.0048; r = −0.7783) were demonstrated.

## 4. Discussion

Macrophages are innate immune cells that act as a fundamental link between the innate system and the adaptive response. They are exceptionally flexible in recognizing and responding to a great variety of stimuli, and their heterogeneity relies on a differential gene expression as a result of the reciprocal interaction with other cells and mediators [28]. This plasticity regarding effector functions, cell surface receptor expression, and cytokine production leads to a differentiation in polarized populations. In the context of an inflammatory environment, the presence of IFN-γ, IL-1R signaling, and TNF-α results in an M1 activation with potent endocytic, phagocytic, and secretory functions, while cytokines such as IL-4, IL-13, IL-10, or TGF-β induce the M2 activation, playing different, even opposite functions. Importantly, while the high plasticity of macrophages allows them to adapt to the different organism’s needs and maintain homeostasis, this functional malleability can also be manipulated and exploited by pathogens, such as viruses and tumor cells, for their own benefit [28]. Tumor and stromal cells in TME produce cytokines, chemokines, and growth factors that recruit macrophages and guide their polarization to an anti-inflammatory, angiogenic, and tumorigenic phenotype known as TAMs, generally with an M2-like phenotype, promoting tumor invasion and metastasis. Given that TAMs infiltration is associated with poor prognosis in cancer patients, reprogramming them toward an antitumor M1 phenotype, inhibiting their recruitment, and suppressing their survival have become possible targets for cancer therapies [29].

On the other hand, EBV-associated lymphomas have been demonstrated to have a higher prevalence in children younger than 10 years old [24], patients whose innate immune system represents the main response to EBV infection. In addition, a particular recruitment of macrophages in pediatric HL TME, which is affected by EBV status, has been described [30,31]. However, little is known about the macrophages’ role in the context of EBV primary and persistent infections in pediatrics. These findings lead us to deepen our studies on the behavior of the most important cytokines in the macrophage polarization direction and their relationship with EBV. As far as we know, this is the first work that characterizes the macrophages’ most relevant polarization cytokine milieu at the periphery compartment in the context of EBV infection and compares it with the tonsil environment, which is the site of viral entrance and replication.

As in many pathologies and inflammatory processes, peripheral blood reflects tissue events; we looked for a connection between serum and tissue cytokine expressions in our patients. Unexpectedly, no correlation was demonstrated for TNF-α, IL-1β, CCL17, or IL-10 when the series was analyzed as a whole, indicating that, at least for those cytokines, the peripheral compartment does not mirror what happens in tissue. Only when infectious status was analyzed separately (PI, HC, and R) was a negative correlation between plasmatic and tissue IL-10 evidenced exclusively in R patients. This preliminary finding may be a signal of a viral influence on the interplay between the periphery and tissue, depending on the infection stages.

The EBV infection has a wide range of presentations, from infectious mononucleosis (IM) with fever and lymphadenopathy primary infection to a silent, persistent infection. The diverse symptomatology regarding infectious phases is attributed to a differential immune response, which can be modulated through cytokine expression, produced by several sources. When viral impact on periphery cytokines (IL-12p70, TNF-α, IL-6, IL-10, IL-1β, CCL17, IL-1Ra, IL-12p40, and IL-23) involved in polarization was evaluated by comparing each cytokine expression among the infectious statuses, even though a higher expression of IL-10 in tissue was previously proved in PI [14], unexpectedly, this increase was not observed in the periphery. Contrary to the report by Hsi et al. on EBV-associated Hodgkin lymphoma, our results did not show an increase in plasmatic IL-10 [32]. A higher expression in patients undergoing reactivation was demonstrated for IL-1β, IL-12p40, and IL-23 pro-inflammatory cytokines. In the same way as in the comparison between peripheral and tissue cytokines, the reactivated patients presented a singular behavior. Moreover, we demonstrated a broader expression of IL-1β in the peripheral blood of patients undergoing reactivation, unlike the downregulation previously described in the context of EBV primary infection [33]. In addition, we previously proved that IL-1β expression in tonsils also increased with lytic antigen expression in R patients [14]. These findings indicate that, in each stage of infection, EBV can modulate peripheral cytokine expression differently, and IL-1β may play a role only in this particular subgroup, in which peripheral IL-1β may reflect tonsillar behavior.

EBV has largely been described to modulate the infected B cell as well as innate and adaptive immune cell functions by regulating intracellular pathways through its latent and lytic proteins. In a previous work, we characterized the tonsil microenvironment in the context of EBV infection in pediatric patients and demonstrated a lower expression of the proinflammatory TNF-α cytokine when Latency II or III viral proteins were expressed [14]. In line with this, in the context of the Latency 0/I antigen expression, our patients showed a broader expression of plasmatic TNF-α, thus reflecting tissular behavior, as well as IL-1Ra. Budiningsih et al. described a correlation of plasmatic TNF-α levels with EBV viral load as a marker for active infection in Indonesian children infected with P. falciparum [34]. In contrast with these results, in the presence and absence of lytic antigen, we observed a higher expression of plasmatic TNF-α in BMRF1-negative patients, confirming previous observations in tissue in R patients [14]. These results suggest that the expression of EBV latent and lytic antigens are related to proinflammatory TNF-α cytokine expression. Given that in this series, PI children mostly displayed restricted EBV antigens [27], these results support the hypothesis that the lack of symptoms in infected children could be related to the decreased expression of this pro-inflammatory cytokine in the context of active viral infection.

We discussed above the key role of cytokines in macrophage polarization, as well as their role as effectors in the macrophages’ response to stimulus. Furthermore, we demonstrated an influence of EBV in the expression of some of the most relevant cytokines involved in polarization. When we analyzed the relation of peripheral cytokine expression and macrophage polarization, we evidenced that CCL17, an important attractant of regulatory T cells, secreted by anti-inflammatory macrophages, was highly expressed in patients that presented an M2 profile at the site of viral infection, suggesting a connection between tonsils and the periphery scenario. Additionally, the inverse correlation of CCL17 at the GC with M2 macrophages exclusively in HC and R may suggest that CCL17 is secreted to the periphery and does not persist in tissue in the context of persistent infection. On the other hand, we previously demonstrated the prevalence of M1 macrophages in an IL-10 regulatory environment triggered by EBV primary infection [14]. Therefore, the positive correlation of IL-23, a cytokine involved in directing polarization to the M1 profile, with CD163, a marker of M2, in HC might evidence a distinctive response of macrophages to stimulus in the context of EBV infection.

In summary, in this study we demonstrated that EBV and its latent and lytic proteins might influence the expressions of several local and peripheral cytokines produced by multiple sources and thereby the macrophage polarization process. Moreover, as most of these findings occurred in the context of viral persistence, we strongly believe that this stage of infection impacts macrophage polarization markers and, ultimately, may influence the development of EBV-associated lymphomas. More studies in this field are needed to better comprehend the process of EBV-associated lymphoma development in children and thereby achieve better cancer-fighting therapies.

## Figures and Tables

**Figure 1 viruses-15-02105-f001:**
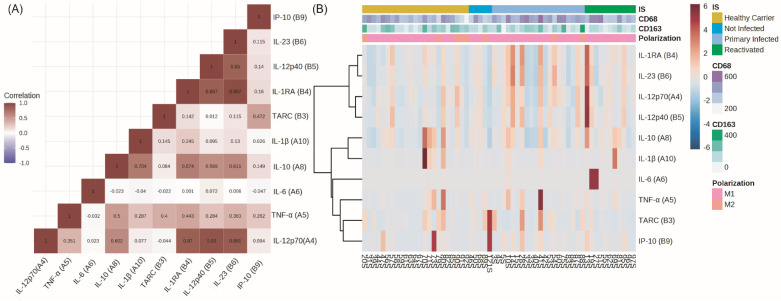
(**A**) Correlation matrix of peripheral cytokines. Numbers inside the box represent the Spearman correlation coefficient (r). The color within each box indicates the value of r, the red color indicates values between 0 and 1 (positive correlation), and the blue color indicates values between 0 and −1 (negative correlation), while the intensity of the color increases as r increases towards values close to |1|. (**B**) Heat map of peripheral cytokine expression (color scale) arranged by infectious status (IS); polarization markers (CD68 and CD163) and polarization profiles (M1 and M2) data are shown at the top. Both graphs show two clusters of cytokines with distinct behaviors. One cluster is compounded by IL-12p40, IL-12p70, IL-23, and, unexpectedly, IL-1RA. On the other hand, the second cluster can be further subdivided into two groups: IL-10 and IL-1β, and IL-6, TNF-α, CCL17 (TARC), and IP-10.

**Figure 2 viruses-15-02105-f002:**
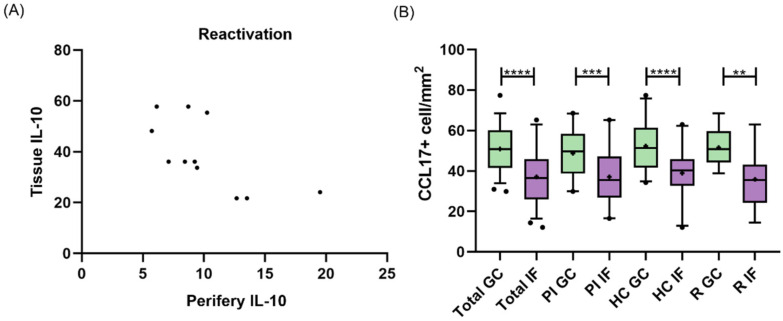
(**A**) Correlation between tissue and peripheral IL-10 expression in reactivated patients. (**B**) Histological distribution of CCL17. Total expression in the germinal center (GC) and interfollicular region (IF) in the entire cohort in primary infected patients (PI), healthy carriers (HC), and patients undergoing reactivation (R). The cross inside the box represent the mean, the line crossing the box represent the median, and error bars indicate the standard deviation. ** *p* < 0.01, *** *p* < 0.001, **** *p* < 0.0001.

**Figure 3 viruses-15-02105-f003:**
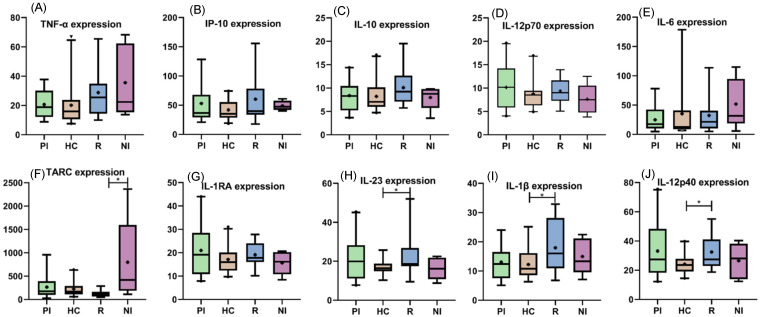
Peripheral cytokine expression among infectious statuses. (**A**) TNF-α. (**B**) IP-10. (**C**) IL-10. (**D**) IL-12p70. (**E**) IL-6. (**F**) TARC. (**G**) IL-1RA. (**H**) IL-23. (**I**) IL-1β. (**J**) IL-12p40. The cross inside the box represents the mean, the line crossing the box represents the median, and error bars indicate the standard deviation; * *p* < 0.05.

**Figure 4 viruses-15-02105-f004:**
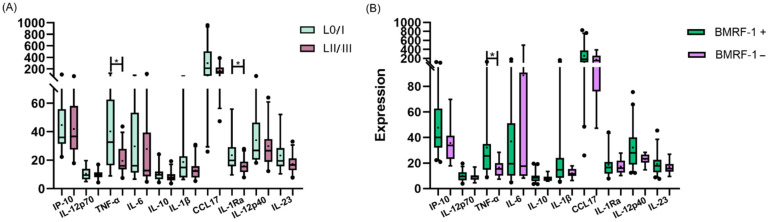
(**A**) Peripheral cytokine expressions in patients expressing Latency 0 and I antigens (L0/I) and Latency (II/III) antigens (LII/III). (**B**) Comparison of peripheral cytokine expressions between patients who were BMRF1+ and BMRF1−. The cross inside the box represents the mean, the line crossing the box represents the median, and error bars indicate the standard deviation. * *p* < 0.05.

**Figure 5 viruses-15-02105-f005:**
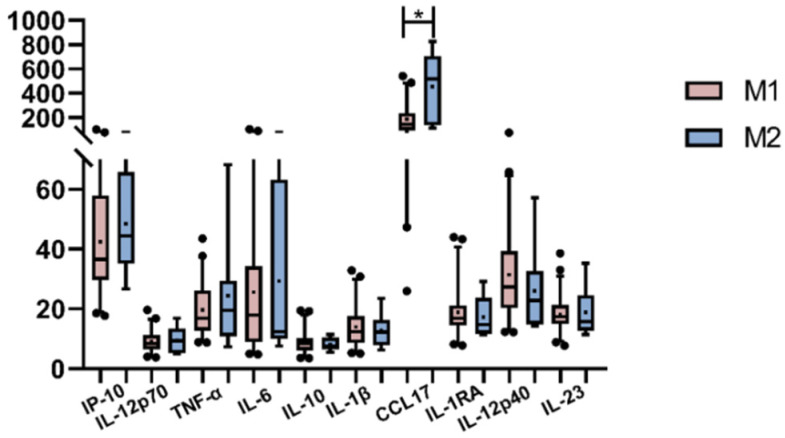
Peripheral cytokine expression in patients with the prevalence of M1 or M2 polarization profiles in tonsils. The cross inside the box represents the mean, the line crossing the box represents the median, and error bars indicate the standard deviation. * *p* < 0.05.

## Data Availability

The datasets generated during and/or analyzed during the current study are available from the corresponding author upon reasonable request.

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
