# Peer review of "EBV Impact in Peripheral Macrophages’ Polarization Cytokines in Pediatric Patients"

_viruses, 2023, doi:10.3390/v15102105_

Round 1
Reviewer 1 Report
The manuscript authored by Agustina Moyano et. al. focused on to better understand the involvement of macrophages in EBV pediatric infection, investigators examined the behavior of peripheral cytokines involved in macrophage polarization and linked it to tissue findings. They investigated the expression of cytokines in tonsils and peripheral blood samples from children at various phases of illness. In tonsils, peripheral cytokines were matched to macrophage polarization markers and viral protein expression. Only in patients undergoing viral reactivation did IL-10 exhibit a negative link between compartments. In reactivation children, there was a greater expression of peripheral IL-1, IL-23, and IL-12p40. Lower expression of local and peripheral TNF- was seen in patients with higher expression of latent and lytic viral proteins. In healthy carrier (HC) patients, IL-23 strongly linked with CD163, and IP-10 positively connected with CD68. These results suggest that EBV may regulate antigen expression in the presence of TNF- and influence peripheral cytokine expression differently depending on the stage of infection. Furthermore, peripheral cytokines may have a role in macrophage polarization in HC. Moreover, peripheral cytokines might have a particular role in macrophages polarization in HC. These findings are quite intriguing, and they could help in understanding the complex role of macrophages in EBV pediatric infection. The study is straightforward, and there are no major objections to publishing the findings.
N/A
Author Response
Response: we thank the reviewer for its comments on our manuscript. Please find attached below the revised manuscript with track changes including all reviewers’ suggestions.

Reviewer 2 Report
Moyano et al. studied the expression of cytokines iaffecting macrophage polarization n peripheral blood samples and tonsils from EBV-infected children, including primary infected individuals, healthy carriers, and children undergoing EBV reactivation. Samples from not infected individuals were studied as well. In paralel, polarization markers of M1 and M2 macrophages were assessed, using immunohistovhemistry, in tonsil specimens, too. The authors did not observe a correlation between serum and tissue cytokine expression, with the exception of IL-10, in a group of children undergoing EBV reactivation. This may indicate, as suggested by the authors, that lytic EBV replication has an effect on the interplay between cytokine levels measured at the periphery and tonsil tissue. This is a well controlled study addressing a multifaceted problem. I would recommend, however, the publication of this manuscript only after a revision.
Minor points:
in line 29, The Epstein-Barr virus… please change to: Epstein-Barr virus…
in line 84, ...IL-1Ra… stands for Interleukin 1 receptor antagonist; in lines 182-183 ...IL-1RA… stands for the same; please unify
in line 224, please amend the title of Figure 3
please provide legends to Supplementary Figure 1
Major points:
in lines 120-123 the authors mention that ”The studied cohort enlarges previous studies [14],… - they refer to, however, on a single publication – please amend or clarify.
In the same sentence, they give numbers of study participants: 20 Primary Infected (PI), 23 Healthy Carriers (HC), 11 undergoing reactivation (R) and 5 not infected (NI) patients – these numbers look similar to the participants in one of their previous studies [14. Moyano, A.; Ferressini Gerpe, N.M.; De Matteo, E.; Preciado, M.V.; Chabay, P. M1 Macrophage Polarization Prevails in 421
Epstein-Barr Virus-Infected Children in an Immunoregulatory Environment. J. Virol. 2022, 96, e0143421.] with the exception of Healthy Carriers (HC); the numper of participants in this group seems to be less, than in the previous study (reference [14]), so the studied cohort does not enlarge the previous study referred to in that particuliar sentence. Please clarify and amend that sentence.
I recommend a revision of this manuscript.

Author Response
Response: all minor points were included at the corrected version. All major changes were included. Concerning the specific sentence about reference n°14, we meant that this study deepens the analysis performed in a previous series, not that the number of patients were increased. In order to clarify this issue, the sentence was amended. Please find attached the track changes version of this manuscript, including the reviewers suggestions.
